

# Investigation of scale interaction between rainfall and ecosystem carbon exchange of Western Himalayan Pine dominated vegetation

Sandipan Mukherjee[1], K Chandra Sekar[1], Priyanka Lohani[1], Kireet Kumar[1], Prabir Patra[2], and Kentaro Ishijima[2]

[1]GB Pant National Institute of Himalayan Environment and Sustainable Development, Kosi-Katarmal, Almora, 263643, India.
[2]RCGC/IACE/ACMPT, Japan Agency for Marine-Earth Science and Technology (JAMSTEC), Yokohama, 236-0001, Japan.

**Correspondence:** S Mukherjee (sandipan@gbpihed.nic.in); PK Patra (prabir@jamstec.go.jp)

**Abstract.**

Forests of the Western Himalaya, India, are impacted by the summer monsoon and winter seasonal rainfall events and associated changes in the meteorology. Here, we assess the scale interactions between observed forest ecosystem fluxes and meteorological parameters, particularly rainfall seasonality and extremes. The scale interactions were investigated using daily

observed fluxes and meteorological parameters of 1080 days of 2014 - 2016 from a *Pinus roxburghii* dominated forest and using wavelet spectral analysis method. The mixed forest of this study was a sink of $CO_2$ having the average NEE -3.21 $gC.m^{-2}.day^{-1}$ for the period of observations. Result of the wavelet coherence analysis from observed data indicated a statistically significant correlation ($> 0.7$ at 95% confidence level) between daily average NEE and daily total rainfall having band periods of 70-120 days, 35-64 days and 60-90 days of monsoon periods of 2014-16, respectively, where rainfall leading to

NEE. Impact of heavy rainfall events of monsoon periods over NEE of the forest patch was found to have average band periods of 4 days; whereas, the winter time heavy rainfall events were having average band periods of 15 days with very high local correlation ($> 0.7$ at 95% confidence level) inferring that ecosystem exchange rate was mirroring rainfall events. Although CASA-GFDE3 model simulated daily NEE values of 2014-15 were found have a low fraction of explained variance (= 0.08) with respect to observations, modeled NEE-rainfall relationship of 2014 was found to corroborate well with the observed pat-

tern; however for 2015, the phase relationship between modeled NEE and rainfall around band period of 100 days was opposite to the flux tower observations. Subsequently, heavy rainfall events and daily average air temperatures were also found to be coherent during monsoon period having wavelet coherences $> 0.8$ indicating a cause and effect relationship between both parameters, and rainfall events were mirroring temperature variations. Therefore, it is anticipated that the *Pinus roxburghii* dominated forest productivity of Western Himalaya, India, are expected to increase even with the increment of heavy rainfall

events in the near future.

## 1   Introduction

The Western Himalaya, India, typically receives two spells of rainfall in a 12-month cycle, one during 122 days of June to September; and the other during 123 days winter months of December to March. The June to September rainfall is associated





to the monsoon circulation and contributes around 700 mm rainfall for the season (Mukherjee et al., 2015) with respect to all India monsoon total rainfall of 850 mm (Parthasarathy et al., 1994), whereas the winter rainfall events are associated to easterly moving Western Disturbances contributing around 180 mm rainfall (Dimri, 2005). The summer monsoon seasons over the Western Himalaya are often associated to heavy rainfall events ($> 75$ mm/day), similarly, the winter rainfall events of Western

Himalaya are often associated to snowfall up to 1200 m elevation. Heavy to extreme rainfall events of summer and winter seasons and associated changes in the atmospheric conditions, such as changes in solar radiation, moisture availability, air and land surface temperature variation etc., significantly force the land and water cycles of the region (Paul et al., 2000; Kothyari et al., 2004; Gabet et al., 2004; Bookhagen et al., 2005). Subsequently, the forced land - water - atmospheric conditions are known to impact forest ecosystem exchange of carbon and water (Cowan, 1977; Farquhar and Sharkey, 1982; Baldocchi, 2008),

such as, air temperature difference between below and above the canopy is found to modulate the forest micro-climate during severe heat stress periods (Renaud and Rebetez, 2009; Renaud et al., 2011); gross primary productivity (GPP) and ecosystem respiration (RE) is observed to be highly impacted by the water and extreme temperature stress (Reichstein et al., 2013; von Buttlar et al., 2018), which affects the net ecosystem exchange (NEE) at the regional scale over the tropics (Patra et al., 2005) ; and extreme precipitation events like typhoons during Asian summer monsoon are known to demonstrate strong coupling

to ecosystem functioning (Kang et al., 2009; Kwon et al., 2010; Hong and Kim, 2011). Himalayan forests and vegetations are no exception and are expected to be impacted by summer and winter time variability in rainfall, particularly heavy rainfall events, and associated changes in the meteorology. Hence, assessments of scale interactions between forest ecosystem fluxes of Western Himalaya and meteorological extremes would further enable us to infer impact of climate change scenarios to carbon sequestration rates.

In spite of substantial forest cover of Uttrakhand, an Indian state of Western Himalaya [the recorderd forest covers approximately 24,414.8 $km^2$ area in which Pine (*Pinus roxburghii*) occupies 3,943.8 $km^2$ resulting 16.1% of total forest area of the state (Rawat et al., 2011)], almost no information is available on the magnitude and scales of coupling between forest ecosystem functioning and seasonal rainfall variability. In fact, continuously monitored data of water and carbon exchange between the forests and atmosphere, and controls of forest ecosystem exchnage (such as temperature, rainfall, humidity, etc.) are so rare

for the forests of Western Himalaya that numerical relationships between carbon uptake and meteorological controls, as developed by Leith (1972) and Law et al. (2002), are not yet established. Subsequently, lack of such routine network measurement of $CO_2$ flux over Indian subcontinent continues to be a constrain for improved estimates of carbon budget (Patra et al., 2011). Moreover, knowledge of intra-seasonal and inter-annual variability of net ecosystem exchange (NEE) of different forest types in the region is also very limited. Only recently, Watham et al. (2014) have reported NEE of a mixed forest of Terai region and

Singh et al. (2014b) have reported interaction of $CO_2/H_2O$ exchange with the light and water use efficiency of a young pine (*Pinus roxburghii*) forest of Garhwal Himalaya, India. Therefore, given the significant lacuna on understanding scale interaction and coupling mechanism between ecosystem exchange and seasonal rainfall distributions of Western Himalaya, this study particularly aims at assessing the role of monsoon and winter seasonal rainfall events in controlling ecosystem functioning using observed ecosystem flux data, observed meteorological data and wavelet analysis. Here, the terminology 'scale interaction'

is used to emphasize the interaction of temporally small to medium scale meteorological events (2 days $\leq$ duration of event $\leq$



256 day) with non-linear sudden changes in ecosystem fluxes. Therefore, assessment of changes in the ecosystem fluxes with respect to small to medium temporal scale changes in meteorological events, particularly rainfall, is primarily one-way, i.e. monsoon climate to forest flux. However, along with the identification of cause and effect relationships between two events, wavelet method, particularly wavelet coherence method used in this study is able to identify phase relationships indicating

precedent or succeeding events with respect to other. Furthermore, efficiency of a carbon cycle model to simulate comparable scale interactions between ecosystem flux and rainfall variability was assessed using wavelet coherence between ERA meteorology driven CASA-GFED3 terrestrial carbon cycle model simulated net ecosystem exchange and daily observed total rainfall. However, no additional statistical validation is carried out between model and observed NEE except for estimation of root mean squared error and bias.

## 10  2  Study area, instrumentation and methods

### 2.1  Site description and instrumentation

Meteorological and $CO_2/H_2O$ flux data were obtained from a field station at Kosi-Katarmal, Almora, India (29°3′N, 79°3′E), at an elevation of 1217 m above mean sea level (Fig. 1) and falls within *Cwa* category of K*ö*ppen climate classification. The field station is situated in the upper Kosi-watershed (Fig. 1a) and within the campus of GB Pant National Institute of Himalayan

Environment and Sustainable Development, Kosi-Katarmal, and exposed to an eastward down-slope with an average inclination of 5-10° towards north-eastern direction. The forest patch surrounding the flux tower was around 27 Ha. About 6.75 Ha area was found to be under natural pine (*Pinus roxburghii* Sarg., 689 ind/Ha) forest, and remaining 20.25 ha forest area was mixed forest type. Significantly dominant tree species within the forest patch excluding *Pinus roxburghii* Sarg. were *Quercus leucotrichophora* A. Camus (220 ind/Ha), *Celtis australis* L. (148 ind/Ha), *Albizia lebbeck* L. Benth (73 ind/Ha) and

*Cedrus deodara* (Roxb. *ex* D. Don) G.Don (50 ind/Ha). The dominant shrubs were *Pyracantha crenulata* (D. Don) M. Roem. (115 ind/Ha). Further, there were two dominant herbaceous species namely, *Cymbopogan iwarancusa* (Roxb.) Schult. (11000 ind/Ha) and *Micromeria biflora* (Buch.-Ham *ex* D. Don) Benth. (7000 ind/Ha). The average crown height of surrounding mixed vegetation was 12 m. A 3-D sonic anemometer (CSAT3, Campbell Scientific Inc., Utah, USA) and an infra red $CO_2/H_2O$ gas analyzer (EC150, Campbell Scientific Inc., Utah, USA), installed at 30 meter height having a measuring frequency of 10 Hz,

were operational with a net radiation sensor (CNR 4, Kipp and Zonnen, Netherlands) and an automatic raingauge (TE525, Campbell Scientific, Utah, USA). Wind speed and direction sensors (HMP45, Vaisala) along with atmospheric temperature and relative humidity sensors were installed at 2, 20 and 32 m heights of the flux tower.

### 2.2  Measurement detail and data processing

Data presented in this study were for the period of 00:00 Hrs of 16 Jan, 2014 to 23:30 Hrs, 30 Dec, 2016. Therefore, a total of

350, 365 and 365 days of observations were reported for 2014-16, respectively. The monsoon months, i.e. June to September of 2014-16, were represented by day number 137-258 for 2014, day number 152-273 for 2015 and day number 153-274 for 2016.




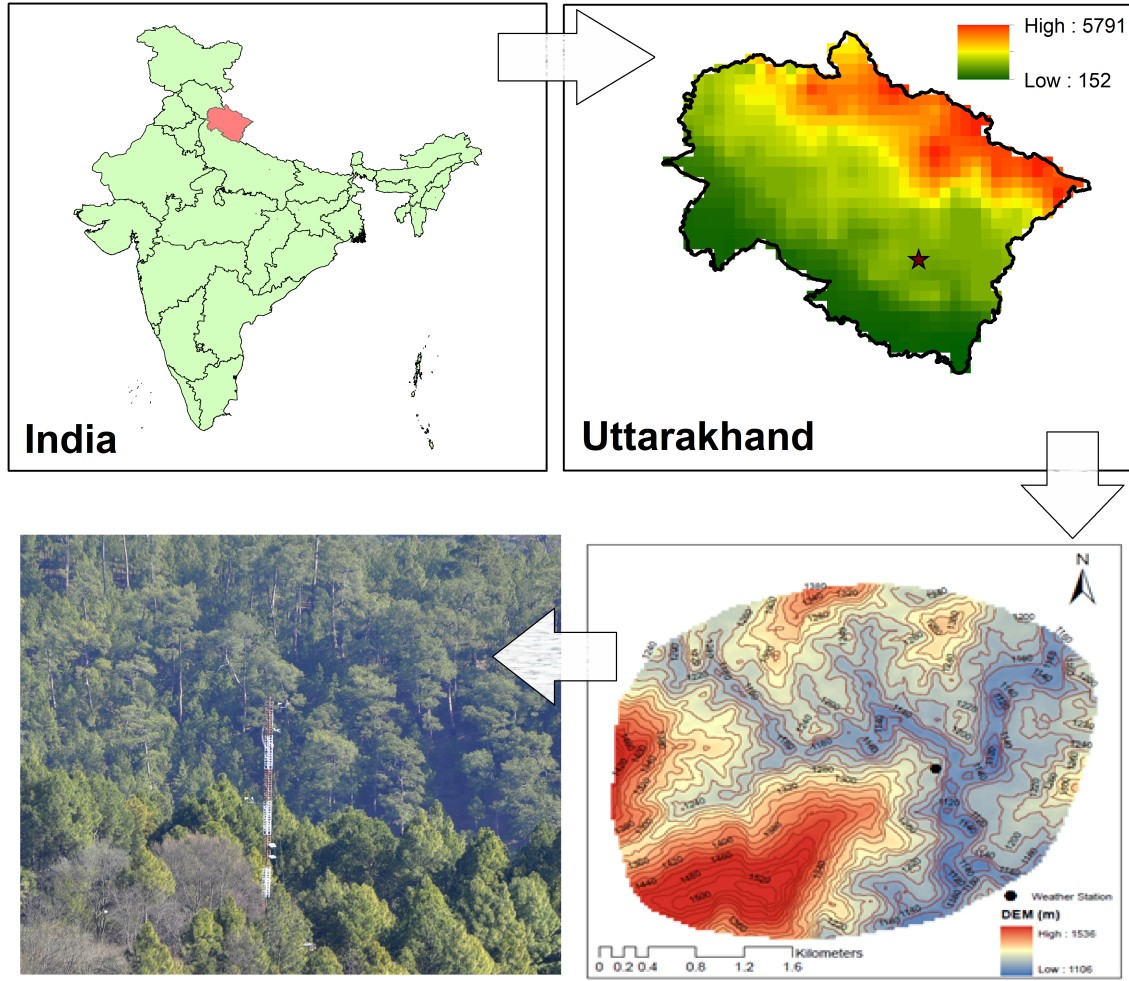

**Figure 1.** Location of the flux tower within the Uttarkhand state of India is represented along with changes in the terrain height, obtained from ETOPO 5 min data, up to a distance of ∼1000 m from the tower location (bottom right panel). The star mark on the upper right panel is showing location of the tower. The bottom left panel is showing the vegetation patch surrounding the weather tower.

The 10 Hz raw data quality assessment was performed using the EddyPro software (version 6.0, LiCor-Bioscience, USA). The quality check assessments were made first, by raw data screening following Vickers and Mahrt (1997). Due to the marginal slope in the underlying terrain, a planar fit correction for tilt in the streamlines was made following the traditional method of Wilczak et al. (2001) where 30° wind sectors were used. A block average detrend method was used and the sonic temperature 5 was corrected for the air temperature using water vapour density (Schotanus et al., 1983) followed by the WPL correction (Webb et al., 1980). The lowpass and highpass filtering effects were addressed following Moncrieff et al. (1997) and Moncrieff et al. (2004), respectively, for the minor deviation in the collocation of sonic anemometer and gas analyser, and the detrending

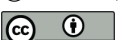



operation. The displacement height ($d$) was computed using the canopy height (12 m) and parameterisation of Foken and Nappo (2008) and 30 min average runs were produced. The 30 min runs were further checked for quality by visual checks, and by removing outliers of a variable $var$, where magnitude of $var < \mu_{var} \pm n\sigma_{var}$. $\mu$ and $\sigma$ were the mean and standard deviation and $n = 3.5$. Furthermore, spike detection methods of Papale et al. (2006) was used to improve data quality. Data gaps

were filled by using the methods of Reichstein et al. (2005). Occasional data gaps for the meteorological parameters, such as vapour pressure deficit (VPD), air temperature, evapotranspiration (ET) were filled using a simple linear interpolation method and rainfall data gaps were filled using long term climatological mean of India Meteorological Department data near Almora district within Uttarakhand state, India. Flux partitioning to gross primary productivity (GPP) and ecosystem respiration (RE) were carried out using the methods Lasslop et al. (2010) and using REddyProc (v 1.1.3). Finally, daily averages of NEE, GPP

and RE were computed for 1080 days of observations. A negative NEE indicates carbon assimilation by the ecosystem.

### 2.3 Terrestrial carbon cycle model

In order to assess efficiency of a carbon cycle model to simulate scale interactions between ecosystem flux and rainfall variability, wavelet coherence between NEE, simulated by ERA meteorology driven CASA-GFED3 terrestrial carbon cycle model (Potter et al., 1993; Randerson et al., 2015), and daily observed total rainfall was investigated. The 3 hourly NEE values were

produced from the CASA-GFED3 monthly values at 0.5-degree spatial resolution using flux downscaling methods of Fisher et al. (2016) where ERA-interim meteorology for short wave radiation and temperature analysis at 6 hourly intervals (00, 06, 12, 18 UT) were filled with 3 hourly forecast fields. The 3 hourly model simulated NEE values were extracted for a grid surrounding the flux tower at Kosi-Katarmal, Almora, India (29°3′N, 79°3′E) for period of 00:00 Hrs of 16 Jan, 2014 to 23:30 Hrs, 31 Dec, 2015. A total of 715 daily average NEE values were produced from the 3-hourly values and wavelet coherence

was estimated using observed daily total rainfall for the same duration.

### 2.4 Wavelet analysis

In order to assess scale interaction between rainfall and ecosystem exchange, continuous wavelet transform (CWT) and cross wavelet transform (XWT) along with wavelet coherence test (WTC) were applied to both met-data and ecosystem exchange data on a daily scale. Wavelet transform is an effective tool to extract localized frequency information in a time scale by

segregating scale contribution of individual events (Katul and Vidakovic, 1996; Salmond, 2005). Therefore, wavelet analysis is extensively used for exploratory data analysis and identification of non-stationarity in a time-series. CWT method is well know for feature extraction from a geophysical time series (Grinsted et al., 2004; Domingues et al., 2005; Weib et al., 2011); similarly, XWT can be considered as a special case of CWT where power spectrum of two time series is analyzed simultaneously. Both CWT and XWT applications of this study were made using numerical formulations of Torrence and Compo (1998)

and details of the numerical formulations are not provided here. The CWT with Morlet wavelet was carried out on the daily average air temp, VPD, NEE, GPP, RE and total rainfall for the entire duration of observations, i.e. for 1080 days using maximum scale of 256 day with 12 sub-octaves per day. Similarly, XWT of daily average NEE and daily average air temp, vapour pressure deficit and total Rainfall was carried out for 1080 days using maximum scale of 256 day with 12 sub-octaves



per day. The wavelet variance at 95% confidence level were estimated with respect to a red noise generated through first order auto-regressive process. The WTC, used to identify phase similarity between two signals, was estimated using Grinsted et al. (2004) where the smoothing operator, which is analogous to correlation coefficient, was kept as 0.6 as described in Weib et al. (2011). Statistically significant wavelet coherence was estimated using Monte-Carlo simulation of first order auto-

regressive process of a background spectra. For XWT and WTC between ERA meteorology driven CASA-GFED3 NEE and observed daily total rainfall, 715 data points of 2014-15 were used with wavelet parameterisation similar to one described above. The wavelet analyses were carried out using scripts of Torrence and Compo (1998) and Grinsted et al. (2004) from http://www.glaciology.net/wavelet-coherence.

## 3   Results and discussion

### 3.1   Turbulence statistics and surface layer meteorology

The total numbers of 30 min observed runs for 2014 to 2016 were 16800, 17520 and 17520, respectively and, a total of 268, 189 and 210 numbers of runs were found to be highly convective ($z/L <$ -10.0); similarly, the highly stable atmospheric conditions ($z/L > $ 10.0) were observed for 4, 8 and 3 numbers of runs, respectively. *Unstable near neutral* atmospheric surface layer conditions (-2.0 $\leq z/L \leq$ -0.5) were found to be dominating throughout 2014 (21.1% of total runs), 2015 (20.9% of total runs)

and 2016 (20.6% of total runs); similarly, *Stable near neutral* atmospheric surface layer conditions (0.5 $\leq z/L \leq$ 2.0) were found to be 9.6, 10.1 and 10.3%, respectively, for 2014-2016 (Fig, 2 a.i-iii). In order to assess impact of surface roughness and terrain on the shear stress, maxima of friction velocity ($u_\star$) was plotted with respect to wind speed and direction for each year (Fig 2 b.i-iii), and it was noted that, irrespective of observation period, the surface roughness and terrain induced stress was minimum ($u_\star \leq$ 0.4 ms$^{-1}$) for the north-easterly sector of the flux tower having approximately 5-10° down-slope in terrain,

whereas high terrain induced shear stress ($u_\star \geq$ 0.7 ms$^{-1}$) was noted for the westerly sector corroborating to up-slope terrain induced change in surface roughness. Relatively homogeneous variation in $u_\star$ (0.4 $\leq u_\star \leq$ 0.7 ms$^{-1}$) and wind speed was noted for the southerly section of the flux tower.

Variability in the surface layer meteorological conditions during observation period, i.e. 00:00 Hrs of 16 Jan, 2014 to 23:30 Hrs, 31 Dec, 2016, was assessed producing diurnal variations in wind speed, wind direction, relative humidity and air tem-

perature measured at 30 m height (Fig. 3). The average wind speed, air temperature, and relative humidity for the observation period were 1.03 ($\pm$0.65) ms$^{-1}$, 16.9 ($\pm$ 7.7) $^\circ C$ and 55.8 ($\pm$ 19.8) %, respectively. It can be noted from Fig. 3 (a, e, i) that, irrespective of seasons, calmest surface layer conditions having wind speed $<$ 0.075 ms$^{-1}$ were between 06:00 Hrs to 11:00 Hrs of a day when turbulent mixing was minimum. However, with the progression of daytime period, formation of convective boundary layer was found to modulate the wind regime. Subsequently, irrespective of seasons, dominant wind direction during

highly convective periods of 09:00 Hrs to 15:00 Hrs were found to be between 0-120°. However, average three year maxima of air temperature (23.6 $^\circ C$) and average three year minima of relative humidity (30.8 %) were observed around 14:30 Hrs.

Since this study is particularly focused on assessing scale interaction between rainfall and ecosystem exchange, details of rainfall variability for the observation period were produced in Table 1. The rainfall events were classified based on India





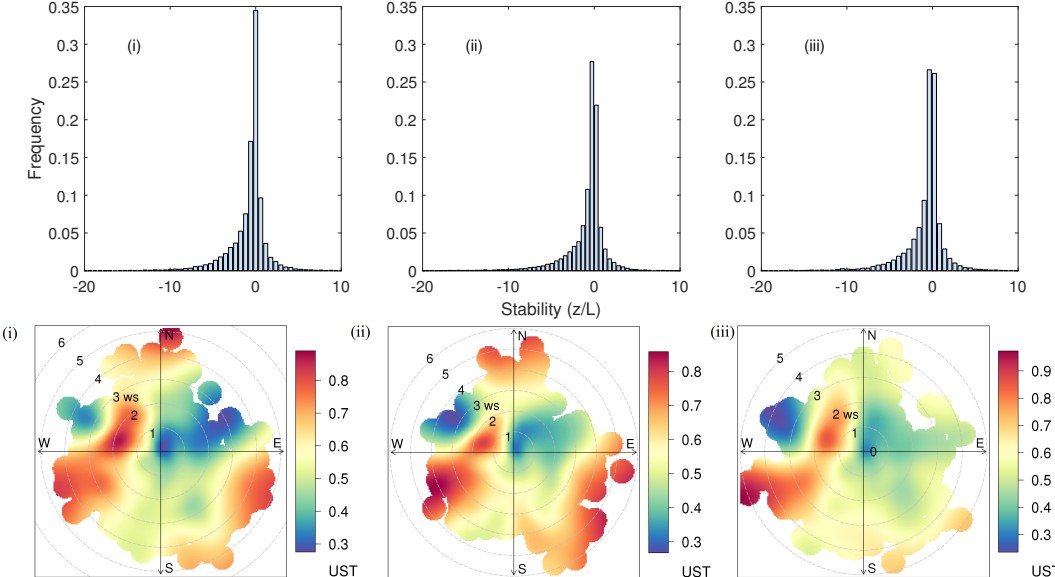

**Figure 2.** Histograms of stability parameter ($z/L$) are represented for (upper panel. i-iii) 2014-2016, respectively, having bin sizes of 0.59, 0.52 and 0.60. Maxima of friction velocity ($u_\star$) are presented with respect to wind speed and direction for (lower panel. i-iii) 2014-2016, respectively.

Meteorological Department [Table 1 of Pattanaik and Rajeevan (2010)]. The total number of rainy days were 109, 111 and 91, respectively, for 2014-2016, of which light rain (daily rainfall < 10 mm) constituted around 76% of total rainfall events. The total amount of rainfall for 2014-16 were 777.5, 744.5 and 703.5 mm, respectively, of which total monsoon period (June-September) rainfall was 592.9, 490.0 and 545.0 mm, respectively; and winter period (January-February, November-December) rainfall was 99.1, 36.3 and 8.4 mm, respectively. It is to be noted that extremely wet winter of 2014 was due to a total of 98.9 mm rainfall of December of that year. In contrary to 2014, a dry winter was noted for 2016 when almost no rainfall events had happened during November and December.

### 3.2 Continuous wavelet transform of meteorological parameters

CWT was carried out to daily time series of average air temperature, average vapour pressure deficit and total rainfall to identify significant feature in the time series, particularly features associated to heavy rainfall events of monsoon and winter period (Fig. 4). The y-axis of Fig. 4 represents periods of spectra in 'Days' whereas the x-axis represent time in days of observations. This is to be noted that the colours of scalogram represent variance of wavelet where blue and yellow represent weak (scale 1/32 and 1/64) and strong (scale 32 and 64) wavelet variance, respectively. Similarly, the shaded area below the thin black line of each scalogram represent the cone of influence where wavelet power drops significantly and the wavelet variance can have considerable artifact.





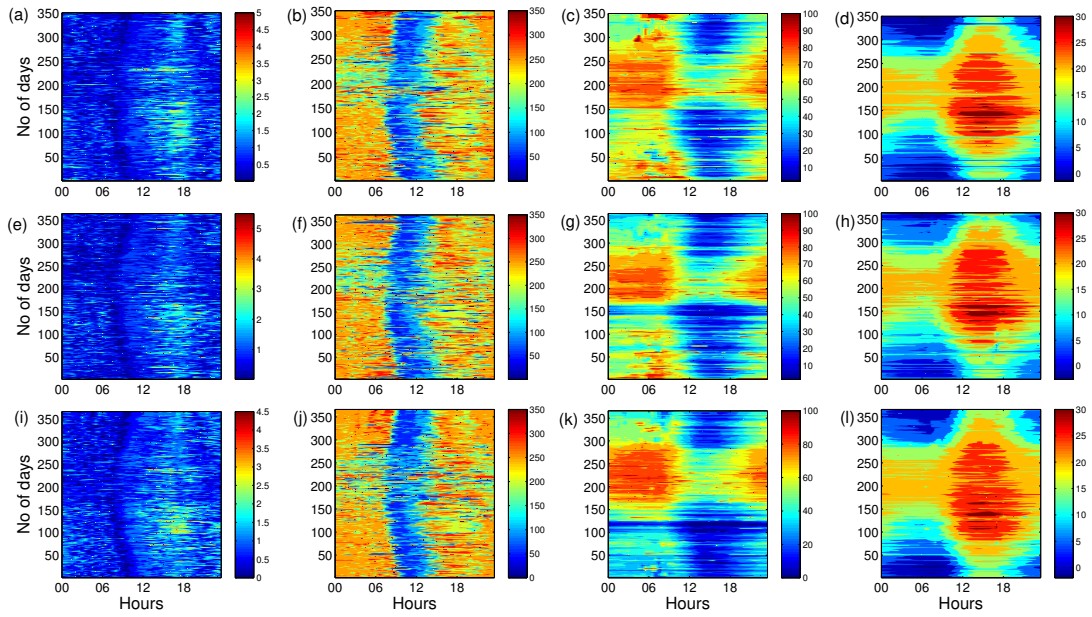

**Figure 3.** Diurnal variations are presented for wind speed (ms$^{-1}$, subplots a, e, i), wind direction ($^{\circ}$, subplots b, f, j), relative humidity (%, subplots c, g, k) and air temperature ($^{\circ}C$, subplots d, h, l) measured at 30 m height. Subplots (a to d), (e to h) and (i to l) are for 2014 to 2016, respectively.

**Table 1.** Annual and monsoon rainfall distributions over the Kosi-Katarmal station are represented based on daily rainfall events.

| Year | Rainfall events | | | | | | | |
|------|-----------------|---|---|---|---|---|---|---|
| | *Light* | | *Moderate* | | *Rather heavy* | | *Heavy* | |
| | No. of days | Avg. rain | No. of days | Avg. rain | No. of days | Avg. rain | No. of days | Avg. rain |
| 2014-Annual | 84 | 2.3 | 16 | 17.6 | 8 | 44.2 | 1 | 121.4 |
| 2015-Annual | 86 | 2.5 | 22 | 17.7 | 2 | 48.9 | 1 | 68 |
| 2016-Annual | 68 | 1.4 | 19 | 17.8 | 2 | 61.7 | 2 | 73.1 |
| 2014-Monsoon | 39 | 2.6 | 10 | 16.9 | 4 | 44.4 | 1 | 121.4 |
| 2015-Monsoon | 40 | 2.9 | 13 | 16.1 | 2 | 48.9 | - | - |
| 2016-Monsoon | 31 | 1.7 | 12 | 18.4 | 2 | 61.7 | - | - |

Now, one can note a significantly high variation in the wavelet power of VPD and rainfall (scales varying between 4-64) but not for air temperature (Fig. 4). This can be corroborated to the fact that the observed daily variations of VPD and rainfall at the present site were much more asymmetric than temperature (asymmetry indices of these parameters were $ind_{rain} = 0.8; ind_{VPD} = 0.3$; and $ind_{airtemp} = 0.05$, asymmetry index was computed as $ind = |S_{sig}/\sigma_{sig}|$, where $S_{sig}$ and $\sigma_{sig}$ is the skewness and standard deviation of a signal $sig$. Dominant signatures (within 95% confidence level) identifiable from spectra





of daily average temperature were 2-8 day band of premonsoon period of 2014, and 2-16 day band of early monsoon period of 2015. However, unlike 2014-15 where pre-monsoon and early monsoon temperature signatures can be corroborated to sustained summer time heating, significantly localized peaks between 2-4 days of premonsoon period were noted for 2016 indicating much smaller heating impact.

The dominant signatures (within 95% confidence level) of VPD were the regular occurrence of 16 day band of premonsoon period occasionally extending to 3rd week of June, and 5-10 day band of premonsoon period of 2014 and 2016. The scalogram of daily average VPD depicts strong signature (scales between 2-16) peaking at the Julian days of 159, 144 and 143 of 2014-16 correlating the driest weather observed during last week of May and first week of June. A high wavelet variance of daily average VPD can also be noted for a 60-100 days scale before every monsoon season of 2014-2016 corroborating to drier
months of March to May.

It can be noted from Fig. 4 subplot (c) that wavelet spectra has well captured the monsoon seasonal rainfall of 2014-2016, where most of the rainfall events have shorter time duration ($< 8$ days). Similar shorter duration rainfall events of winter and pre-monsoon seasons are also clearly represented in the rainfall power spectra. However, notable rainfall events having significantly high wavelet power spectra for monsoon periods were (i) a 16 day rainfall event of 2014 when a total of 393.3 m
of rainfall was observed over the station with around 121.4 mm rainfall was observed on 19th June; (ii) a 8 day rainfall event of 25 June - 2 July, 2015 when total rainfall of 207.4 mm was recorded; and (iii) 69.1 mm and 77.1 mm rainfall of 1st and 17 July, 2016 resulting two peaks in the power spectra. Similarly, notable rainfall events having significantly high wavelet power spectra for winter periods were (i) 36.8 and 31.6 mm rainfalls of 14 and 28 Feb, 2014; (ii) 57.1 mm rainfall of 15 Dec, 2014 and (iii) a 30.2 mm rainfall event of 9 March, 2015. However, no significant rainfall event was observed during the winters of
20   2016.

### 3.3   Continuous wavelet transform of observed ecosystem fluxes

The Himalayan Pine dominated vegetations are significantly different than the other mixed forests of the region with respect to low percentage of individuals for brief and synchronous leaf flushing and flowering (Ralhan et al., 1985). The first and second leaf flushing of Pine dominated vegetations occur during early summer (March and April), and rainy season (mid-
June to mid-September) (Ralhan et al., 1985). These continuous flushing events can be corroborated to enhanced growth rate of the Pine, and subsequently, expected to increase the carbon sequestration rate. On the other hand, maximum leaf drop is recorded during winter season (mid-December to mid-March) when growth rate subsides; consequently, carbon sequestration rate is expected to decrease. Along with this rationale of ecosystem flux variability of the study area, interaction of fluxes and meteorological controls are elaborated. The observed month-wise variations of daily average NEE, GPP and RE are presented
in Fig. 5. The mixed forest patch of this study was found to be a sink of $CO_2$ as the average NEE for the period of observations were found to be -3.21 $gC.m^{-2}.day^{-1}$ whereas a similar value of -3.13 $gC.m^{-2}.day^{-1}$ was reported by Watham et al. (2014) for a mixed forest of Himalayan foothill of Uttarakhand, India. However, the average NEE of this present forest was lower than the temperate deciduous and conifer forests (5-10 $gC.m^{-2}.day^{-1}$) reported by Wofsy et al. (1993) and Greco and Baldocchi (1996). The annual carbon uptake of the forest under consideration is higher (approx. 1172 $gC.m^{-2}$) than the global average





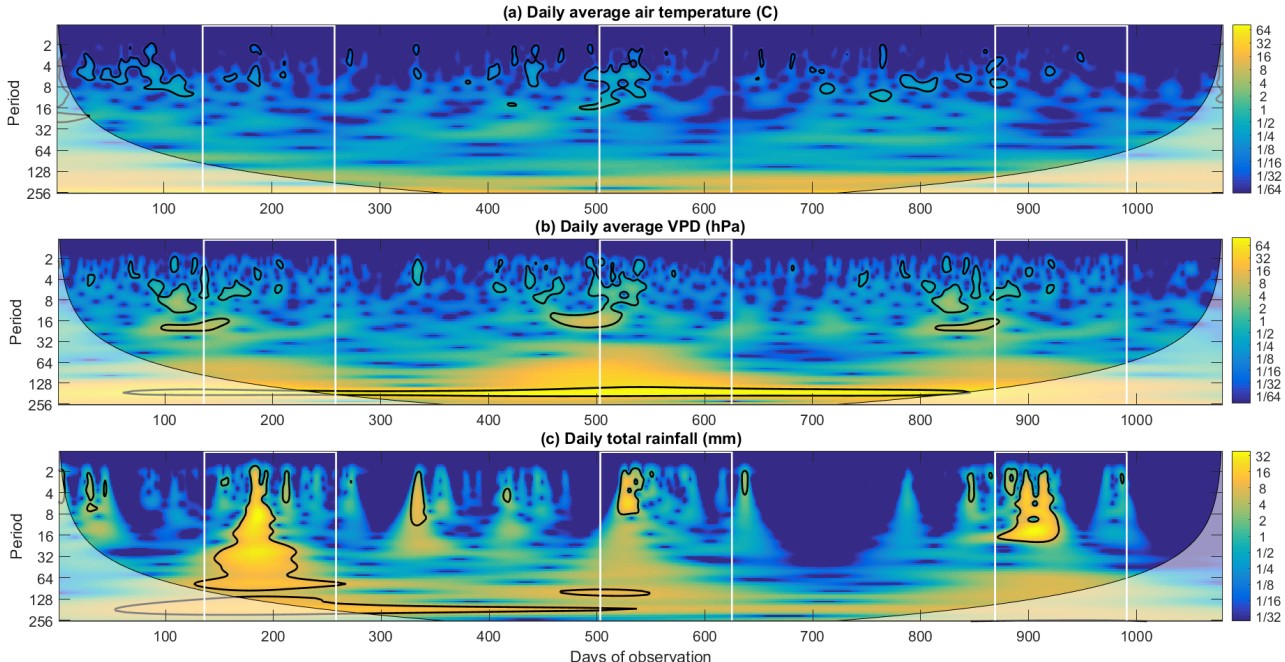

**Figure 4.** CWT of daily average air temperature, VPD and daily total rainfall. The white boxes are showing three monsoon seasons of 2014-2016. 95% confidence level with respect to red noise is shown with black contour lines. The cone of influence is shown with blurred shade.

carbon uptake of 183±270 $gC.m^{-2}$ as estimated by Baldocchi (2008) from 506 different sites of global biomes. However, significantly high carbon uptakes ($> 800$ $gC.m^{-2}$) by some individual forests are not unusual (Tang et al., 2012; Novick et al., 2015). The nine month average estimate of NEE around -3.13 $gC.m^{-2}.day^{-1}$ from a forest of Himalayan foothill by Watham et al. (2014) also indicates a significantly high annual uptake of approx. 1142 $gC.m^{-2}$. The apparent reason for such high carbon sequestration rate of this current vegetation under consideration is the low value of RE as the three year average GPP/RE was around 4.5. Further, the low RE value can be attributed to the rather acidic nature of top-soil having relatively smaller thickness under Pine stands of Himalaya restricting the microbial activity. Such low total respiration from the acidic soils of *Pinus roxburghii* vegetations of Himalaya are already reported in Joshi et al. (1991). The average GPP value of the site (5.6 $gC.m^{-2}.day^{-1}$) was found to be marginally lower than the average GPP of temperate coniferous forests (5.7-6.9 $gC.m^{-2}.day^{-1}$, Falge et al. (2002)). Irrespective of seasons and observation years, the daily average maximum and minimum values of NEE were between 2.61 and -10.93 $\mu.mol.m^{-2}.s^{-1}$, whereas the maximum of GPP and RE were 12.29 and 5.34 $\mu.mol.m^{-2}.s^{-1}$, respectively.

As expected, the monsoon months were having the highest daily average carbon assimilation rate (NEE Jul-Sep = -3.72, -4.27, -3.59 $\mu.mol.m^{-2}.s^{-1}$) along with enhanced evapo-transpiration (average ET = 2.1 mm/day), corroborating to the growing season, than winter months (NEE Nov-Jan = -2.45, -2.03, -2.94 $\mu.mol.m^{-2}.s^{-1}$, average ET = 0.9 mm/day) when leaf



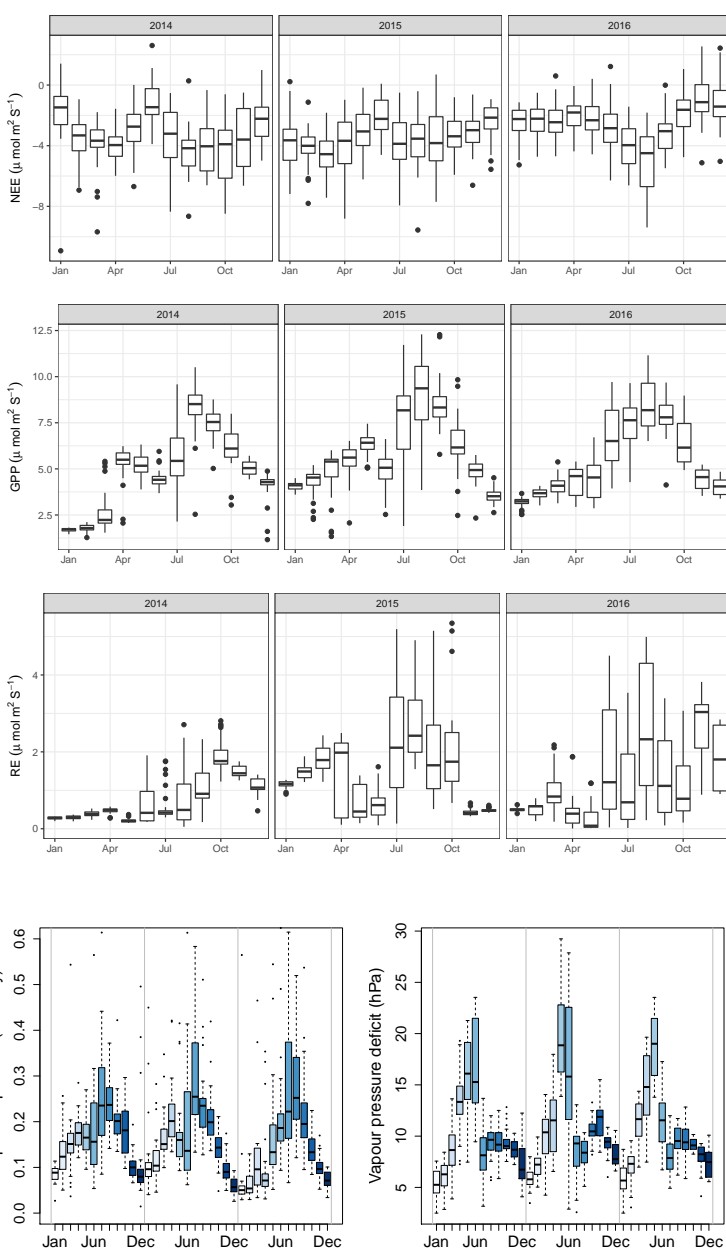

**Figure 5.** Month-wise boxplots of daily average NEE, GPP and RE, respectively, for 2014-2016. The lower panel shows month-wise boxplots of daily average evapo-transpiration and vapour pressure deficit.





abscission was at peak. Moreover, this seasonal changes of productivity can also be corroborated to changes in the green leaf area index which was earlier reported to be highest during late July-early August ( 2.5 $m^2 m^{-2}$) and lowest during January (0.8 $m^2 m^{-2}$) by Singh et al. (2014a) of a *Pinus roxburghii* forest of Uttarakhand, India.

A modest enhancement in the carbon assimilation rate was also observed for the pre-monsoon months of March and April, particularly for 2014 and 2015, than the winter seasons (NEE Mar-Apr = -3.61 and -3.31 $\mu.mol.m^{-2}.s^{-1}$) due to (i) new sprouting of needles and leaves in *Pinus roxburghii* and *Quercus leucotrichophora*, two dominant species surrounding the flux tower, and (ii) faster growth of herbaceous species like *Cymbopogan iwarancusa*. Such enhancement of carbon assimilation rate for the month of March and April with respect to February and May is also reflected in the GPP values of 2014-15 (Fig. 5). This signature was not apparent in 2016 possibly due to significant number of occurrences of forest fire surrounding the forest patch suppressing natural growth of herbaceous species and delayed sprouting in *Pinus roxburghii* and *Quercus leucotrichophora*. However, the carbon assimilation rate subsided with the increasing air temperature (average values varied between 20-25°C) and vapour pressure deficit (average values varied between 15-20 hPa) of May and June of 2014-15.

The CWTs of daily average NEE, GPP and RE are provided in Fig. 6. One can note that unlike GPP and RE, power spectra of NEE were evenly distributed for all three monsoon periods having considerably high wavelength variance (scales 8-16) for shorter time duration ($\leq$ 4 days). However, wavelet decomposition patterns for GPP and RE were found to be similar and can be corroborated to monsoon precipitation. Statistically significant moderately strong wavelet band (scales 1-8) of GPP were observed for each monsoon season having periods of 2-8 days. On a time scale, these dominant signatures were noted for 40-60 days of monsoon season broadly corroborating to average total number of light (45 days) rainy days of a monsoon season over Western Himalaya (Mukherjee et al., 2016). Premonsoon period dominant bands (< 10 days) of GPP were also noted for 2014 and 2015.

Similar to GPP, dominant power spectra of RE (significant band of 8-32 days and 4-16 days) were noted for monsoon seasons of 2015-2016 and premonsoon season of 2016, respectively. The dominant and statistically significant power of RE of 2016 can be independently corroborated to two factors, (i) related to substantial changes in the autotrophic and/ heterotrophic respiration, and/ (ii) related to the estimation of RE itself using methods of Lasslop et al. (2010). Since ecosystem respiration is represented as sum of autotrophic and heterotrophic respiration, estimates of soil microbial activity, soil temperature and soil moisture contents are necessary to comprehend substantial seasonal variation of RE (Falge et al., 2002). However, due to observational constrains, estimates of these parameters could not be linked here. Similarly, RE, as estimated using day time flux partitioning methods of Lasslop et al. (2010), can have substantial artifact due to improper fitting of rectangular hyperbolic light–response curve as mentioned by Falge et al. (2001) and/ significant accumulation of random error in the NEE values. A separate analysis is, therefore, required to properly identify rationale for such variation of RE during 2016.

### 3.4 Cross wavelet transform of selected meteorological parameters and ecosystem fluxes

The XWT and WTC were carried out between rainfall-air temperature and rainfall-VPD (Fig. 7) to assess phase interactions. The XWT spectra between air temperature and rainfall indicates a common power having bands between 2-30 days and 2-16 days at 95% confidence level of monsoon period of 2014 and 2015. A common power around 2-16 days period was also




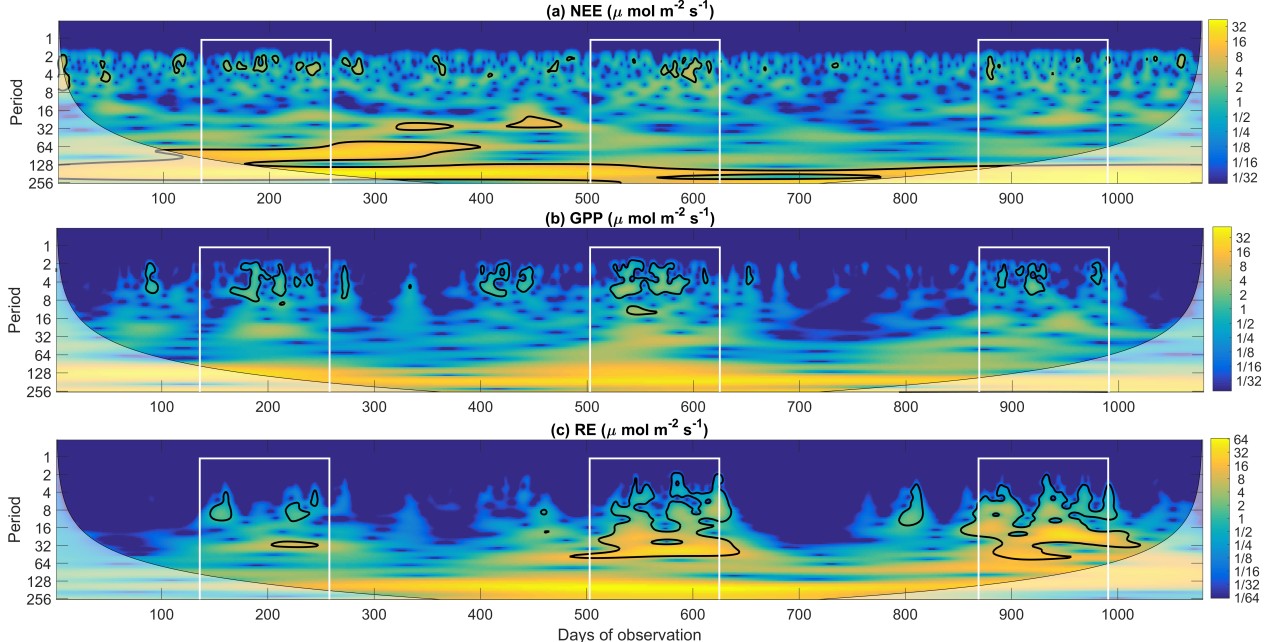

**Figure 6.** CWT of daily average NEE, GPP and RE. The white boxes are showing three monsoon seasons of 2014-2016. 95% confidence level with respect to red noise is shown with black contour lines. The cone of influence is shown with blurred shade.

intermittently observable for 2016, however, the interactions were anti-phased signifying that rainfall events were mirroring temperature variations. Subsequently, a statistically significant (within 95% confidence level) wavelet coherence (>0.75) was observed between both parameters.

One can note that the XWT spectra between VPD and rainfall indicates a common power having bands between 2-30 days and 2-16 days at 95% confidence level of monsoon period of 2014 and 2015 with a common power around 2-16 days period intermittently observable for 2016. However, unlike the air temperature - rainfall XWT spectra, no discernible phase interactions were noted between these two parameters. Therefore, it can be inferred that no consistent cause-effect relationship exists between VPD and rainfall events during early monsoon period. However, statistically significant (within 95% confidence level) anti-phased wavelet coherence (>0.75) could be noted between VPD and rainfall during monsoon retrieval stages having bands of 8-16 days, 8-32 days and 4-16 days of 2014-16, respectively. Similar to the observations of Hong and Kim (2011), it was noted that for the early monsoon 16 days rainfall band was leading the temperature enhancement (downward arrow). Several observable multi-phased relationships between rainfall and temperature could be linked to observations of Berg et al. (2009) who noted that rainfall- temperature relationships change with changing rainfall types such as, orographic, convective and stratiform. With respect to assessment of scale interactions between extreme rainfall events and air temperatures, anti-phased 3-days, 4 days and 8-days bands with wavelet coherences > 0.8 were observed for 19 June, 2014, 25 June - 2 July, 2015, and 1 July - 17 July, 2016 rainfall events indicating a cause and effect relationship between air temperature and rainfall.



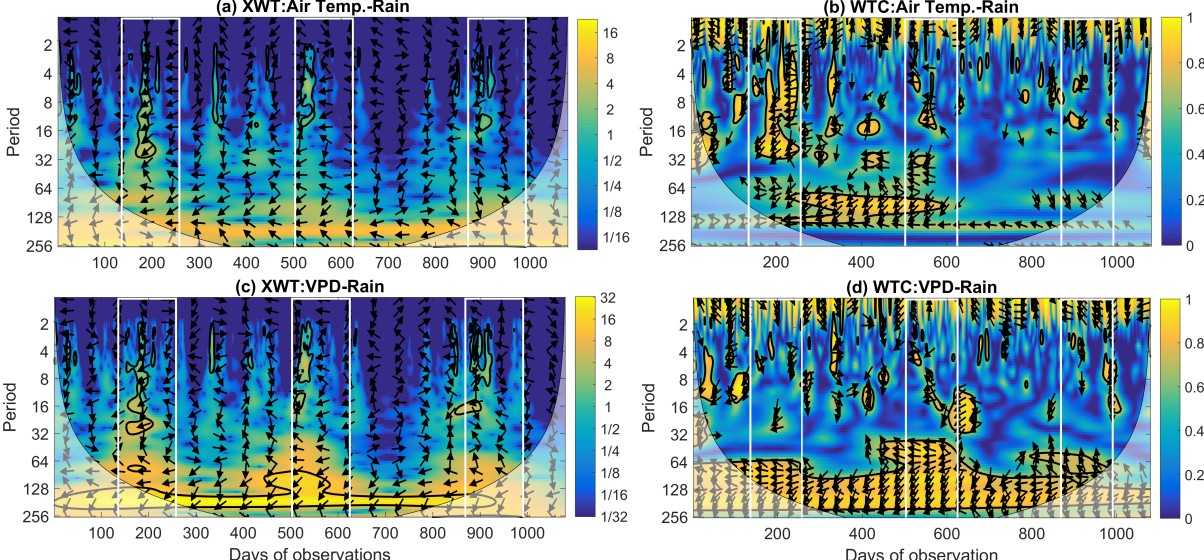

**Figure 7.** Cross wavelet (subplots a, c) and squared wavelet coherence (subplots b, d) of daily average temperature with daily total rainfall (subplots a, c) and daily average VPD with daily total rainfall (subplots b, d) are presented. The white boxes are showing three monsoon seasons of 2014-2016. The 95% confidence level with respect to red noise is shown with black contour lines. The cone of influence is shown with blurred shade. Relative phase angles are represented with arrows; right-arrow represents in-phase; left-arrow represents anti-phase; 90° down-arrow represents rainfall leading air temperature and VPD.

The common power and local correlation distributions between daily average NEE, GPP, RE and daily total rainfall are presented in Fig. 8. The monsoon seasonal common powers between rainfall and all three fluxes can be easily noted from Fig. 8 (a, c, e) with periods varying significantly between 2-64 days having intermittently phase-locked relationship. However, the wavelet coherence spectra were indicating statistically significant correlations (WTC > 0.7) between NEE and rainfall having band periods of 70-120 days, 35-64 days and 60-90 days of monsoon periods of 2014-16, respectively and, rainfall leading to NEE (down ward arrow). The GPP-rainfall relationships, similar to Hong and Kim (2011), were also discernible for monsoon periods of 2014 and monsoon retreat periods of 2015 indicating an overall enhancement of productivity during monsoon period as corroborated earlier to vegetation growth and highest green leaf area index of *Pinus roxburghii*. However, no coherent relationships were noted between RE and rainfall during monsoon seasons.

The impact of heavy rainfall events of monsoon periods [i.e. 19 Jun, 2014 (121.4 mm), 25 Jun - 2 Jul, 2015 (207.4 mm) and 1-17 Jul, 2016 (146.2 mm)] on NEE of the forest patch was found to have small band periods, on average 4 days; whereas, the winter time heavy rainfall events around 15 Dec, 2014 (57.1 mm) and 9 Mar, 2015 (30.2 mm) were found to have an average band period of 15 days with very high local correlation (> 0.7) inferring that NEE was mirroring rainfall events. Such extended impact of rainfall on growth can be corroborated to lower evaporative loss of water from forest floor during winter periods that increases water retention of top soil layer and enhances growth of herbaceous species.





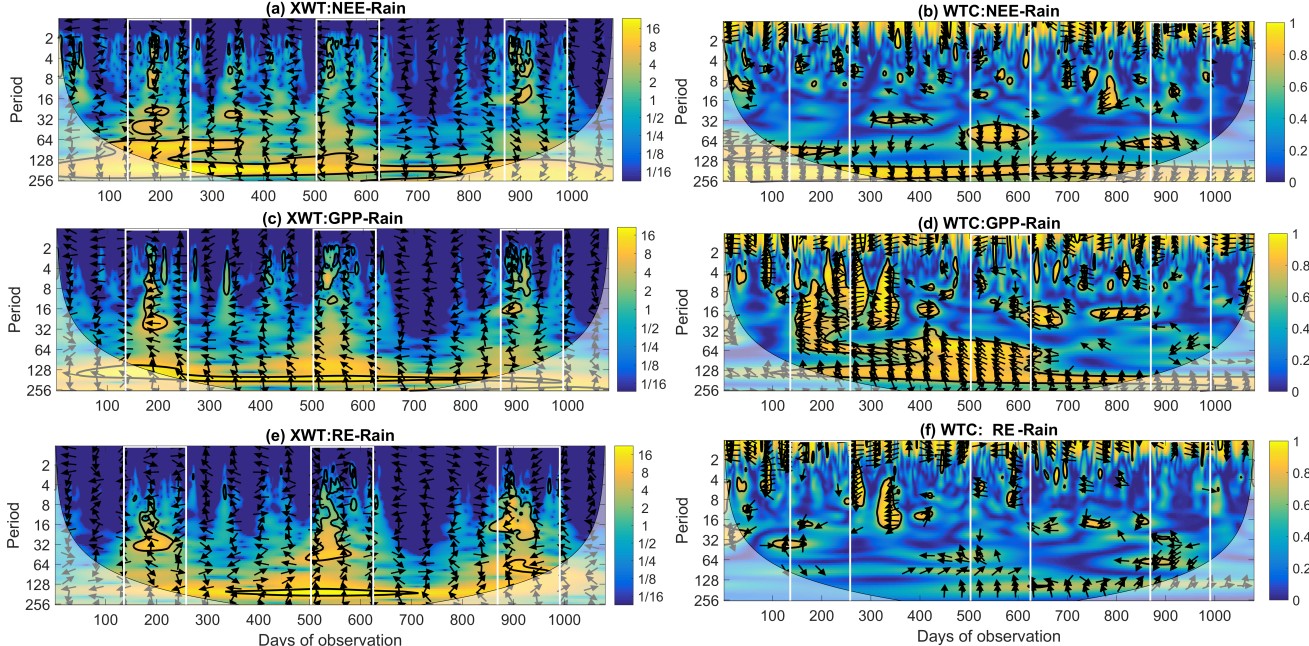

**Figure 8.** Cross wavelet (subplots a, c, e) and squared wavelet coherence (subplots b, d, f) of daily average NEE, GPP and RE with daily total rainfall. The white boxes are showing three monsoon seasons of 2014-2016. 95% confidence level with respect to red noise is shown with black contour lines. The cone of influence is shown with blurred shade.

## 3.5 Cross wavelet transform of rainfall and model simulated net ecosystem exchange

A total of 715 daily average NEE values of 16 Jan, 2014 to 31 Dec, 2015, obtained from ERA meteorology driven CASA-GFED3 terrestrial ecosystem model were used to assess model efficiency to capture NEE-rainfall scale interactions. When the model simulations were compared with the observations (Fig. 9), an average model bias of -3.08 $gCm^2day^{-1}$ was noted. The root mean squared error was minimum for the month of June (1.6 $gCm^2day^{-1}$) and was significantly high for July and August (3.8 and 3.7 $gCm^2day^{-1}$, respectively) due to rainfall induced variability. The fraction of explained variance ($FEV = var_m/var_o$, $var_{o/m}$ represents variance of observed and model simulated data) for both 2014 and 2015 was found to be 0.08 represents a generic under prediction of NEE variability by the model. Similar generic poor prediction of NEE by ecosystem model has been reported by Morales et al. (2005), typically for insufficient representation of water stress effect.

However, to assess small scale interactions between ERA meteorology driven CASA-GFED3 model NEE and observed daily total rainfall, XWT and WTC were produced using data for the period of 16 January, 2014 to 31 December, 2015 (Fig. 10). Similar to the observed data, common powers between rainfall and NEE can be noted from Fig. 10 (a) with periods varying significantly between 2-64 days having intermittently phase-locked relationship particularly for 2014. For 2015, the monsoon period common powers between rainfall and NEE were observed for periods < 8 days and around 100 days band. The wavelet coherence spectra were indicating statistically significant correlations (WTC > 0.5) between NEE and rainfall



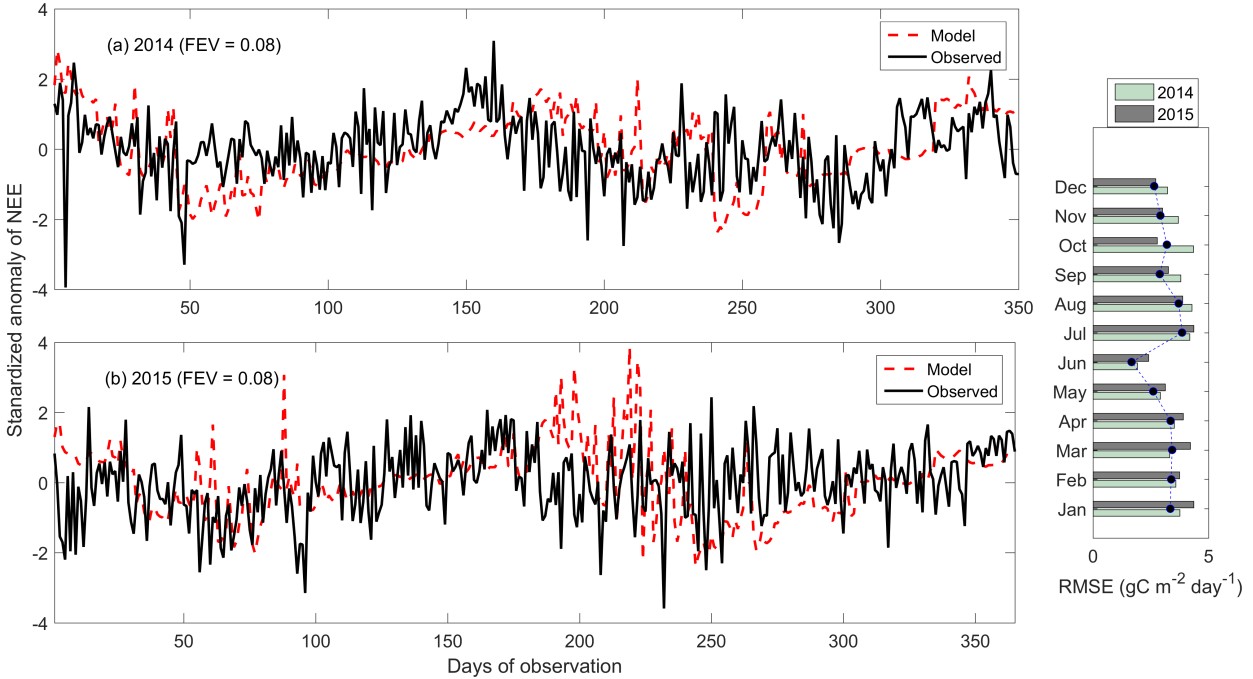

**Figure 9.** Comparison between model simulated and observed standardized anomaly of NEE (($NEE - \overline{NEE})/\sigma_{NEE}$, $\sigma_{NEE}$ is standard deviation of NEE) of 2014-15. The monthly root mean squared errors between model simulated NEE and observations are presented in the right panel. Blue circles with dotted lines are presenting model bias estimate as $|\overline{NEE_{model}} - \overline{NEE_{observe}}|$.

having band periods of 50-60 days of monsoon period of 2014 where rainfall leading to NEE (down ward arrow). Although this NEE-rainfall relationship of 2014 was found to corroborate the observed pattern as indicated earlier; for 2015, the phase relationship between NEE and rainfall around band period of 100 days was found to be opposite to the flux tower observation. In terms of relationship between model predicted NEE and heavy rainfall events of monsoon periods [i.e. 19 Jun, 2014 (121.4

5    mm) and 25 Jun - 2 Jul, 2015 (207.4 mm)], no significant feature was identified for the 19 June event, however, for the 25 Jun - 2 Jul, 2015 rainfall event model NEE was found to have small band periods of 5 day. The winter time heavy rainfall event around 15 Dec, 2014 (57.1 mm) was found to enhance NEE with very high local correlation (> 0.7) having band period of 10 days. Therefore, on a wider perspective, the ERA meteorology driven CASA-GFED3 NEE values were mostly found to have similar scale interactions with rainfall variability as was noted with flux tower measurement.

10  **4    Conclusions**

Approximately 60% land use of Western Himalayan state of Uttarakhand, India is forest, and *Pinus roxburghii* is one of the dominant species found in the forests of Uttarakhand between 800-2000 m elevation. However, ecosystem responses of these *Pinus roxburghii* dominated temperate forests to changes in the meteorological conditions are not yet reported. Therefore, this




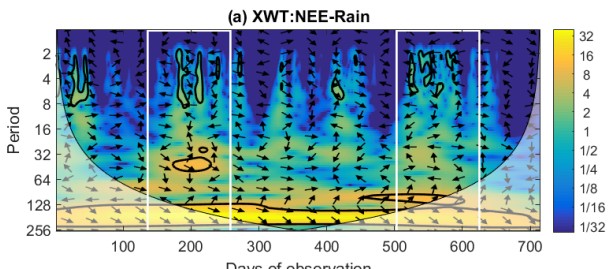
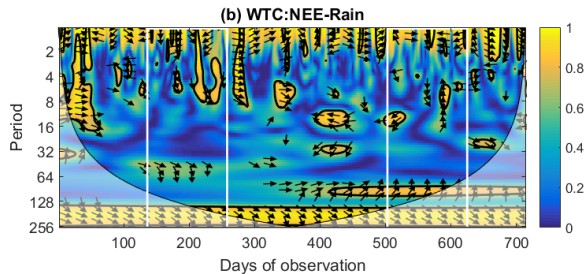

**Figure 10.** Cross wavelet (subplots a) and squared wavelet coherence (subplots b) of daily average NEE from ERA meteorology driven CASA-GFED3 model with observed daily total rainfall for the duration of 2014-2015. The white boxes are showing two monsoon seasons of 2014-2015. 95% confidence level with respect to red noise is shown with black contour lines. The cone of influence is shown with blurred shade.

current study is aimed at assessing the scale interactions between ecosystem fluxes (NEE, GPP and RE) and meteorological parameters, specially seasonal variation of rainfall and heavy to extreme rainfall events. The scale interactions between ecosystem fluxes and rainfall were investigated using daily observed ecosystem fluxes and meteorological parameters of 1080 days of 2014 - 2016 from a *Pinus roxburghii* dominated forest and using wavelet method. Moreover, NEE from a terrestrial ecosystem model was also used to assess efficiency of carbon cycle model to simulate comparable scale interactions with rainfall. Significant findings of this study are summarized as follows:

- The mixed forest patch of this study was found to be a sink of $CO_2$ having the average NEE -3.21 $gC.m^{-2}.day^{-1}$ for the period of observations; and the monsoon months were having the highest daily average carbon assimilation rate (NEE Jul-Sep = -3.72, -4.27, -3.59 $\mu.mol.m^{-2}.s^{-1}$; average ET = 2.1 mm/day) with respect to the winter months having lowest average NEE (NEE Nov-Jan = -2.45, -2.03, -2.94 $\mu.mol.m^{-2}.s^{-1}$, average ET = 0.9 mm/day) when leaf abscission was at peak.

- The scalogram of daily average VPD observed over the site depicted a high wavelet variance of 60-100 days scale before every monsoon season of 2014-2016. Similarly, notable monsoon heavy rainfall events having high wavelet power spectra (Scale > 4) for monsoon periods were (i) a 16 day rainfall event of 2014; (ii) a 8 day rainfall event of 25 June - 2 July, 2015; and (iii) 69.1 mm and 77.1 mm rainfall of 1st and 17 July, 2016. The scalogram of daily total rainfall also depicted high wavelet power spectra (Scale > 4) for winter rainfall events of (i) 36.8 and 31.6 mm rainfalls of 14 and 28 Feb, 2014; (ii) 57.1 mm rainfall of 15 Dec, 2014 and (iii) a 30.2 mm rainfall event of 9 Mar, 2015.

- The cross wavelet spectra between VPD and rainfall indicated a common power having bands between 2-30 days and 2-16 days at 95% confidence level of monsoon periods of 2014 and 2015; a similar common power around 2-16 days period was intermittently observable for 2016. However, no consistent cause-effect relationship could be established between VPD and rainfall events of monsoon period as no significant wavelet coherence was observed.





- The extreme rainfall events and air temperatures were found to be coherent during monsoon period and anti-phased 3-days, 4 days and 8-days bands with wavelet coherences > 0.8 were observed indicating a cause and effect relationship between both parameters where rainfall events were mirroring temperature variations.

- The monsoon seasonal spectral common powers between rainfall and all three fluxes were noted in the cross wavelet spectra. Statistically significant correlations (WTC > 0.7) were observed between NEE and rainfall having band periods of 70-120 days, 35-64 days and 60-90 days of monsoon periods of 2014-16, respectively, and rainfall leading to NEE. Although CASA-GFDE3 model simulated daily NEE values of 2014-15 were found have a low FEV with respect to observations, NEE-rainfall relationship of 2014 was found to corroborate well with the observed pattern; although for 2015, the phase relationship between NEE and rainfall around band period of 100 days was opposite to the flux tower observation. The GPP-rainfall relationships from flux tower measurement, similar to Hong and Kim (2011), were observed for monsoon periods of 2014 and monsoon retreat periods of 2015 indicating an overall enhancement of carbon sequestration rate during monsoon period.

- The impact of heavy rainfall events of monsoon periods on the observed NEE of the forest patch was found to have small band periods, on average 4 days; whereas, the winter time heavy rainfall events were found to have an average band period of 15 days with very high local correlation (> 0.7) signifying that enhancement of ecosystem exchange rate was mirroring rainfall events.

The overarching finding of this article indicate that Indian summer monsoon rainfall events are primary control of *Pinus roxburghii* dominated forest growth of Western Himalaya, India, and monsoon period heavy rainfall events can accelerate forest growth for an approximate lead period of a week. However, heavy rainfall events of winter season can also be a stimulator of instantaneous growth having approximate lead period of up to couple of weeks. Therefore, it can be anticipated that climatological enhancement in rainfall events over Western Himalaya under a changing climate scenario could enhance the forest productivity and carbon sequestration rate.

*Data availability.* Daily flux and met data are available with S Mukherjee and further data sharing needs approval from competent authority of GBPNIHESD, Almora, India

*Competing interests.* No competing interests are present

*Acknowledgements.* The flux tower of Kosi-Katarmal, Almora, India was established through a research collaboration between GBPNI-HESD and CSIR-CMMACS, Bengaluru, India. S. Mukherjee and P. Lohani acknowledge MoES, Govt. of India for a research grant:



MOES/16/25/2014-RDEAS for high frequency data analysis. Mr Ashutosh Tiwari, GBPNIHESD, Almora, India is acknowledged for the digital elevation model of Uttarakhand, India.





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
