# Peer review of "Investigation of scale interaction between rainfall and ecosystem carbon exchange of Western Himalayan Pine dominated vegetation"

_Biogeosciences, 2018_

## Referee Comment (RC1) · Anonymous Referee #1 · 14 Sep 2018

Review comments on 'Investigation of scale interaction between rainfall and ecosystem carbon exchange of Western Himalayan Pine dominated vegetation' by Mukherjee et al.

Please see winter seasonal rainfall to Indian winter monsoon. There are three things mentioned scale interaction, seasonality and extremes. It seems authors don't have understanding of the same. What about orographic forcings over the Himalayas. Over Himalayas most of the precipitation is primarily controlled by orographic forcings. What are the changes in meteorology? It is repeatedly mentioned but not looked into and answered. There are many complex jargons are used in the introduction with lesser

relevance to link with which makes objective of the paper less defined and more complex to understand as what authors are inclined to mention about. ERA-interim forced model CASA-GFED3 is used to assess terrestrial carbon cycle. No sensitivity of the model and comparison with corresponding observations are made with to comment on models suitability. Point, Kosi Katarmal, study is linked with large scale model and fields- that is not advisable. A simple wavelet analysis is forced to justify the objective. Different bands highlighted using wavelet is than linked to the point flux observation. There is lot of lacunae is introduced in such analysis and thus corresponding results. Where is the scale interactions, meteorological changes, etc. finally?

I recommend rejection of the paper based on above main comments. I have not commented and seen this paper from physical processes point of view.

---

## Referee Comment (RC2) · Anonymous Referee #2 · 18 Sep 2018

Investigation of scale interaction between rainfall and ecosystem carbon exchange of Western Himalayan Pine dominated vegetation

Study tries to show the potential effect of Monsoon and winter rainfall over the carbon exchange in an Himalayan pine forest. Eddy Covariance method is used to derived Net Ecosystem Exchange at ecosystem scale during a period of three years. Authors used wavelet analysis to identify time periods of effect between rainfall and other meteorological variables, as well NEE. Authors explored the effect of rainfall patterns with modeled NEE fluxes at "regional" scale with the use of a global product. The topic in general is interesting (Monsoon rainfall and NEE), however the goal of the article is not

clear and not easy to understand until the conclusion. There are not clear hypotheses. Authors assumed that carbon fluxes and productivity are controlled by rainfall patterns, however they also described and showed the potential and combined effect of other biophysical factors. There is an overuse of wavelet analysis and potentially many results can be summarized with words, instead to show many graphical sources. There is a mix of elements in the article (e.g., some methods are described in the results apart and in opposite way). A depth and comprehensible discussion is needed. Some conclusions are faraway of the results showed in the article.

Specific comments:

- Why did the authors decide to use CASA-GFED3 to simulate the NEE? The use of this source is not well justified in the introduction and it makes unclear if the authors wanted to upscale the observed NEE with the modeled NEE. In addition, they did not validate potential relationships between modeled and observed NEE. May be is not necessary the use of this source if the authors make clear the goal of the story. - In the measurement detail and data processing, authors did not say what is fetch filter of their EC tower and how potentially can be affected by the topography. This is relevant because after, authors explain the use of a global product and its unknow how spatially are related the tower fetch with the spatial resolution of the global product. - There is a detailed description about the graphs derived from wavelet analysis, however discussion and references are needed around. For example, why did the authors consider the current meteorological variables in the study and why did they not include others (e.g., radiance, PPFD, etc) that could also explain the forest carbon exchange?, Why during winter rainfall in some years does precipitation affect the productivity of the forest and why others years not? Why the modeled NEE (global product) used is not every time related with the observed NEE? The global product is working properly for this ecosystem? What is the goal to include a global product in the study? - Figure 1 and its caption need improvements. Not scale in the first two maps, not legend, not grid. - The main characters of the story are showing until figure 5, potentially authors

could show the graphs of Figure 5 after Figure 2. - Almost all the captions of the figures are not well explained, and many information of the captions are explained in the main body of the article. It is necessary to improve the Figures captions. - In all the wavelet analysis instead to use DOY in the x-axe, better use regular format date (MM/YYYY)

---

## Author Comment (AC1) · 24 Sep 2018

We appreciate the anonymous reviewer for reviewing the article. With respect to the comment made on 'winter rainfall' this is to emphasize that the rainfall time series used in this study were for the period of 16 Jan, 2014 to 30 Dec, 2016 which include winter seasonal rainfall observation over the site. Furthermore, this is to emphasize that this manuscript is not focused on assessing climatology or dynamics of winter rainfall and/or orographic effect on rainfall over Himalaya as substantial research publications are available for winter rainfall and orographic impact on winter and summer monsoon rainfall over Himalaya (Dimri, 2009, Yadav et al, 2010,Dimri, 2013a, Dimri, 2013b, Yadav et al, 2013, Mukherjee et al, 2016); rather, impact of rainfall seasonality along with changes in the vapour pressure deficit and air temperature, referred in this draft as 'associated meteorology', were assessed on NEE using wavelet. Moreover, in spite of using wavelet to justify objectives of this article, the article is developed around multi-dimensional application of various wavelet methods; and similar application of wavelet methods can be found in Kwon et al, 2010. The rationale for using the ERA forced CASA-GFED3 model was to test efficiency of a carbon cycle model to simulate comparable scale interactions between ecosystem flux and rainfall variability, and that is explicitly mentioned in line number 5, pp-3 of the draft. Similarly, 'scale interaction' terminology is also categorically explained in line no 34-35, pp-2 and subsequently, bands and periods representing various scales are sufficiently elaborated in the results section which may have been over-sighted by the reviewer. However, we do agree that instead of CASA-GFED3 model, a site-specific model could be used to improve understanding of the interactions between rainfall and ecosystem fluxes. Unfortunately, we do not have access to any finer spatial resolution global/regional ecosystem models that could produce carbon exchanges at sub-daily time intervals. Further, as noted by the reviewer, if the lacuna in introduction and analysis section is highlighted/elaborated by the reviewer, could be addressed.

---

## Short Comment (SC1) · 30 Sep 2018

This article explores the mutual relationship between rainfall, air temperature, and VPD and $CO_2$ exchange by a Pine forest over the Western Himalayan foothills in India using wavelet analysis technique. The authors have established the cause-effect relationship between rainfall and ecosystem $CO_2$ exchange using wavelet coherence and cross-spectra. Overall, this article is well written and addresses a problem less explored in the scientific literature so far. It should be accepted after revisions. Following are my comments to improve the paper.

1. Why only the 3-hour scale was considered while downscaling the CASA-GFD3

[Figure]

NEE output which was subsequently used for wavelet coherence with the measured NEE? Is there any natural process associated with this scale that the authors want to probe specifically?

2. In Figures 4, 6 and 7 different scales are used in different panels (a, b, c) such as, in Fig. 4a and 4b wavelet power vary between [1/64,64] and in Fig. 4c between [1/32,32]. These should be uniform for better understanding.

3. The colour scale used in the lowest panel of Fig. 5 should be explained in the figure caption. 4. Why is such a strong seasonality observed in the LAI, between 0.8 $m^2.m^{-2}$ in January and 2.5 $m^2.m^{-2}$ in late July-early August, given the fact that the forest studied here is an evergreen coniferous one?

5. The authors show that the carbon assimilation rate subsided with the air temperature and VPD. But as we know the NEE increases initially with air temperature and VPD and decreases later at very high values with air temperature and VPD, after stomatal closure. The authors need to address this point.

6. Why is the NEE CWT markedly different in 2014 compared to the other two years?

7. Can the authors provide the uncertainty estimates in their NEE, GPP, and RE budget?

8. We know that the land-surface temperature gradient during the pre-monsoon and monsoon drives the Indian summer monsoon. However, the authors state that the early monsoon 16 days rainfall band precedes the temperature enhancement. What is the physical mechanism behind this?

9. How is the RMS error in NEE attributed to the rainfall variability?

10. The authors have used the term 'phase-locked' multiple times in the text to explain the results without much explanation. It should be defined and explained clearly.

11. There are many typos and grammatical mistakes in the manuscript such as "….and controls of forest ecosystem exchnage" in Sec. 1, "…The fraction of explained variance (…) for both 2014 and 2015 was found to be 0.08 represents a generic under prediction of NEE variability by the model." in Sec. 3.5 etc. This list is not exhaustive. These should be carefully examined and corrected.

---

## Author Comment (AC2) · 26 Oct 2018

Final Response: Title: Investigation of scale interaction between rainfall and ecosystem carbon exchange of Western Himalayan Pine dominated vegetation Author(s): Sandipan Mukherjee et al. MS No.: bg-2018-299 MS Type: Research article

Reply to the comments of Reviewer - 1: Thank you very much for reviewing the article. However, due to the lack of detailed comments and suggestions we are unable to make major revisions to the manuscript, although a number of significant modifications are proposed to be made to account for the comments and suggestions from Reviewer#2. Some of these proposed modifications might be useful to clarify some of your con-

cerns. With respect to the comment made on 'winter rainfall' this is to emphasize that the rainfall time series used in this study was for the period of 16 Jan, 2014 to 30 Dec, 2016 which include winter seasonal rainfall observation over the site. Furthermore, this is to elaborate that this manuscript is not primarily focused on assessing climatology or dynamics of winter rainfall and/or orographic effect on rainfall over Himalaya, as substantial research publications are available for winter rainfall and/ orographic impact on winter and summer monsoon rainfall (Dimri, 2009, Yadav et al., 2010, Dimri, 2013a, Dimri, 2013b, Yadav et al., 2013, Mukherjee et al., 2016) rather, impact of rainfall seasonality along with changes in the vapour pressure deficit and air temperature, referred in this draft as 'associated meteorology', were assessed on NEE using wavelet. Moreover, in spite of using wavelet to justify objectives, the article is developed around multi-dimensional application of various wavelet methods; and similar application of wavelet methods can be found in Kwon et al, 2010. The rationale for using the ERA forced CASA-GFED3 model was to test efficiency of a carbon cycle model to simulate comparable scale interactions between ecosystem flux and rainfall variability, and that is explicitly mentioned in line number 5, pp-3 of the draft. Similarly, 'scale interaction' terminology is also categorically explained in line no 34-35, pp-2 and subsequently, bands and periods representing various scales are sufficiently elaborated in the results section. However, we do agree that instead of CASA-GFED3 model, a site-specific model could be used to improve understanding of the interactions between rainfall and ecosystem fluxes. Unfortunately, we do not have access to any finer spatial resolution global/regional ecosystem models that could produce carbon exchanges at sub-daily time intervals. Further, as noted by the reviewer, the lacuna in introduction and in result with respect to rationale and use of CASA-GFED3 model could be elaborated if a revised manuscript is considered.

Reply to the comments of Reviewer - 2: We appreciate the helpful comments and suggestions by anonymous reviewer on the article. With respect to the comment made on 'Goal of the article with underlying hypothesis' this is to emphasize that the overarching aim of this article was to assess interaction between rainfall (both monsoon and winter

seasons) and ecosystem exchange of a Pine dominated forest of Western Himalaya. Subsequently, cause and effect relationship between two important meteorological parameters (i.e. air temp, VPD) and rainfall were also assessed. The study was initiated with the rationale that the western Indian Himalayan Pine dominated mixed forest patches, which encompasses almost 60% land use of the Uttarakhand state of India, are expected to respond disparately to different spells of rainfall during summer and winter which is not yet quantified. Therefore, the particular aim of this study was to quantify this cause and effect relationships between NEE and rainfall using observed data. Moreover, the sub-daily scale global model product was used to assess similar rainfall and ecosystem exchange interactions which can be generalized for the entire Pine dominated forests of Western Indian Himalayan region such as the one reported in Hong and Kim (Global Change Biology,17, 2011). With respect to the comments on 'Over use of wavelet analysis', we have included only those wavelet diagrams (i.e continuous wavelength and cross wavelength) of parameters which were providing significant signatures. Therefore, according to the authors, wavelet analyses included in the draft were optimum and necessary. However, the figure for cross wavelet interaction and wavelet coherence between rainfall-VPD and rainfall-air temperature (Figure 7) could be removed and only the results could be highlighted in a revised draft. The 'use of CASA-GFED3 model and goal of using such global model' is primarily inspired by convenience as we are aware of the coarse resolution of the model to represent the observational site but that is the data we had at our disposal and not many ecosystem model results are available at sub-daily time resolution. Moreover, as indicated by the reviewer, site specific verification of global model products implies a better up-scaling of carbon source-sink distribution, which could be elaborated in the revised draft emphasizing role of validation of global model products with observations for enhancing model accuracy and acceptance of model products for better carbon budget estimation. With respect to the comments made on the 'Fetch area of the mast with role of topography', this is to elaborate that a detailed footprint analysis was not included in the draft, rather variation in the wind shear was represented as a function of wind

speed. However, the theoretical cumulative footprint function (FPc), as estimated using the footprint model of Hsieh et al. (2000) for each 30 minute observations, indicated 80% of the fluxes were originating within 1.06 km of the measurement tower irrespective of the wind direction and atmospheric surface layer stability ($\zeta$ = z-d/L, where z is measurement height, d is displacement height and L is Obukhov length). When the footprint fluxes were partitioned with respect to dominant wind direction, 80% of the fluxes were found to be originating within a radius of 1.06 km for NE wind (0-90o), 0.94 km for SE wind (90-150o) and 1.18 km for NW wind (270-355o) regimes (Figure A1). For convective/unstable periods ($\zeta$< 0), the footprint fetch was found to vary between 40-300 m signifying most of the fluxes were contributed by the surrounding forest patches, whereas, for stable atmosphere ($\zeta$ > 0) maximum footprint fetch was as much as 2400 m, irrespective of wind direction. Further, this is to be noted that the footprint fetch estimation was made under the assumption of horizontally homogeneity of surface sources which was not entirely satisfied by the underlying conditions of the current terrain, hence a separate error estimation for source area identification will be needed which is out of the scope of this draft. Further, we acknowledge that the current manuscript lacks 'discussion on radiance and PPFD' due to non inclusion on these parameters for wavelet analysis, as we have tried to particularly focus on rainfall variability. Comments related to 'Figure captions', 'Change in figure spacing', 'Occasional typos' and 'More detailed discussion of results' could be incorporated in a revised draft.

Reply to the comments of Short Comment - 1: Pramit Deb Burman, IITM , India With respect to comments on the '3-hourly CASA-GFED3 model product' this is to emphasize that the 3 hourly data was converted to daily average values, afterwards, wavelet analysis was carried out. As indicated above, not many ecosystem model results are available at sub-daily time resolution, hence, the CASA-GFED3 data was used. The scale difference in Figs – 4, 6 and 7 are due to inclusion of all statistically significant signatures of different parameters. Hence, a forced equal wavelet scale would either include statistically insignificant information (i.e. inclusion of 1/64 in case of 1/32) or removal of significant information (i.e. restriction of scale up to 1/32 where valid signatures are present in 1/64). The color scale of Fig-5 can be explained in a revised draft. For the detail LAI information Singh et al. (2014a) is referred. The comment on 'relationships between NEE-VPD and NEE-air temperature' is not appropriate as no direct wavelet based relationships between these parameters were derived, rather VPD-rain and air temperature-rain relationships were highlighted to physically correlate meteorological parameters during different seasons and scales. Similarly, as indicated in the draft, power spectra of NEE were evenly distributed for all the three monsoon periods. Since, carbon budget estimation was not the objective of the study, no detail uncertainty and error estimation was carried out. The generic understanding of continental scale temperature gradient and subsequent monsoon rainfall is a regional scale phenomenon within the central and peninsular India. The same monsoonal trough does not propagate much beyond the foothills of Himalaya. Therefore, localized convection and moist adiabatic lifting of air parcel play significant role for precipitation within Himalaya even during monsoon. The temperature enhancement following rainfall, particularly during premonsoon periods, can be corroborated to sufficient latent heat release after rainfall events. This information could be incorporated in the revised draft. The phase-lock conditions in cross wavelet analysis represent periods when two events are having similar oscillation. The details mathematical explanations of wavelet methods are not provided in the draft as the same could be found in Grinsted et al. (2004) and Weib et al. (2011). Similarly, comments related to 'Figure captions', 'Change in figure spacing', 'Occasional typos' could be incorporated in a revised draft.
* * *
[Figure]

**Fig. 1.** Subplot (a) indicates Cumulative footprint function (FPc) with respect to three dominant wind regimes; and subplot (b) indicates variation of footprint fetch (m) with respect to atmospheric stability

---

## Author Comment (AC3) · 29 Oct 2018

Reply to the comments of Short Comment - 1: Pramit Deb Burman, IITM , India

With respect to comments on the '3-hourly CASA-GFED3 model product' this is to emphasize that the 3 hourly data was converted to daily average values, afterwards, wavelet analysis was carried out. As indicated above, the CASA-GFED3 data we had at our disposal and not many ecosystem model results are available at sub-daily time resolution.

The scale difference in Figs – 4, 6 and 7 are due to inclusion of all statistically significant

signatures of different parameters. Hence, a forced equaling scaling during production of images would either include statistically insignificant information (i.e inclusion of 1/64 in case of 1/32) or removal of significant information (i.e. restriction of scale up to 1/32 where valid signatures are present in 1/64).

The color scale of Fig-5 can be explained in a revised draft. For the detail LAI information Singh et al (2014a) is referred. The comment on relationships between NEE-VPD and NEE-air temp. seems not correct as no direct wavelet based relationships between these parameters were derived, rather VPD-rain and air temp.-rain relationships were highlighted to physically correlate meteorological parameters during different seasons and scales. Similarly, as indicated in the draft, power spectra of NEE were evenly distributed for all the three monsoon periods. Since, carbon budget estimation was not the objective of the study, no detail uncertainty and error estimation was carried out.

The generic understanding of continental scale temperature gradient and subsequent monsoon rainfall is a regional scale phenomenon within the central and peninsular India. The same monsoonal trough does not propagate much beyond the foothills of Himalaya. Therefore localized convection and moist adiabatic lifting of air parcel plays a significant role for precipitation within Himalaya even during monsoon. The temperature enhancement following rainfall, particularly during premonsoon periods, can be corroborated to sufficient latent heat release after rainfall events. The phase-lock conditions in cross wavelet analysis represent those periods when two events are having similar oscillation. The details mathematical explanations of wavelet methods are not provided in the draft to avoid repetitions. Similarly, comments related to 'Figure captions', 'Change in figure spacing', 'Occasional typos' and 'More detailed discussion of results' could be incorporated in a revised draft.